# Methyl P-Coumarate Ameliorates the Inflammatory Response in Activated-Airway Epithelial Cells and Mice with Allergic Asthma

**DOI:** 10.3390/ijms232314909

**Published:** 2022-11-28

**Authors:** Ji-Won Park, Jinseon Choi, Juhyun Lee, Jin-Mi Park, Seong-Man Kim, Jae-Hong Min, Da-Yun Seo, Soo-Hyeon Goo, Ju-Hee Kim, Ok-Kyoung Kwon, Kihoon Lee, Kyung-Seop Ahn, Sei-Ryang Oh, Jae-Won Lee

**Affiliations:** 1Natural Medicine Research Center, Korea Research Institute of Bioscience and Biotechnology, Cheongju 28116, Republic of Korea; 2College of Pharmacy, Chungbuk National University, Cheongju 28160, Republic of Korea; 3College of Pharmacy, Chungnam National University, Daejeon 34134, Republic of Korea; 4Laboratory Animal Resources Division, Toxicological Evaluation and Research Department, National Institute of Food and Drug Safety Evaluation, Ministry of Food and Drug Safety, Osong Health Technology Administration Complex, Cheongju 28159, Republic of Korea; 5Natural Product Central Bank, Korea Research Institute of Bioscience and Biotechnology, Cheonju 28116, Republic of Korea; 6Laboratory Animal Resource Center, Korea Research Institute of Bioscience and Biotechnology, Cheongju 28116, Republic of Korea

**Keywords:** asthma, methyl p-coumarate, Th2 cytokines, eosinophil, NF-κB

## Abstract

Methyl p-coumarate (methyl p-hydroxycinnamate) (MH) is a natural compound found in a variety of plants. In the present study, we evaluated the ameliorative effects of MH on airway inflammation in an experimental model of allergic asthma (AA). In this in vitro study, MH was found to exert anti-inflammatory activity on PMA-stimulated A549 airway epithelial cells by suppressing the secretion of IL-6, IL-8, MCP-1, and ICAM-1. In addition, MH exerted an inhibitory effect not only on NF-κB (p-NF-κB and p-IκB) and AP-1 (p-c-Fos and p-c-Jun) activation but also on A549 cell and EOL-1 cell (eosinophil cell lines) adhesion. In LPS-stimulated RAW264.7 macrophages, MH had an inhibitory effect on TNF-α, IL-1β, IL-6, and MCP-1. The results from in vivo study revealed that the increases in eosinophils/Th2 cytokines/MCP-1 in the bronchoalveolar lavage fluid (BALF) and IgE in the serum of OVA-induced mice with AA were effectively inhibited by MH administration. MH also exerted a reductive effect on the immune cell influx, mucus secretion, and iNOS/COX-2 expression in the lungs of mice with AA. The effects of MH were accompanied by the inactivation of NF-κB. Collectively, the findings of the present study indicated that MH attenuates airway inflammation in mice with AA, suggesting its potential as an adjuvant in asthma therapy.

## 1. Introduction

The prevalence of allergic asthma (AA) is increasing globally [1,2]. The sustained airway inflammation induced by allergic sensitization is a major pathophysiological characteristic of AA [3]. Airway epithelial cell-derived cytokines, chemokines, and adhesion molecules (IL-6, IL-8, MCP-1, and ICAM-1) are associated with the development of AA [4,5]. Macrophage-released cytokines and chemokines (TNF-α, IL-1β, IL-6, and MCP-1) affect airway inflammation in asthma pathogenesis [6,7]. Th2 cytokines (IL-4, -5, and -13) and eosinophils play an important role in the development of airway inflammation and mucus secretion in subjects with AA [8,9]. This increase in mucus secretion is a feature of AA and is related to airflow limitations [10,11]. It has also been reported that the increase in the expression of inducible nitric oxide (NO) synthase [iNOS] may affect mucus secretion by inducing NO production in subjects with AA [12,13]. The inhibition of cyclooxygenase-2 (COX-2) has been shown to improve asthmatic symptoms in a clinical study [14] and in mice with AA [15]. The increased activation of NF-κB causes bronchial inflammation, inducing cytokine/chemokine/mediator expression in an experimental models of AA [16,17,18]. Phorbol 12-myristate 13-acetate (PMA) has been used in in vitro asthma studies as it is known to promote the expression of cytokines/chemokines/adhesion molecules and the activation of NF-κB/AP-1 in A549 airway cells [15,19]. Ovalbumin (OVA) has been used in in vivo asthma studies as it induces bronchial inflammation by accelerating eosinophil influx, Th2 cytokine/IgE production, mucus secretion, iNOS/COX-2 expression, and NF-κB activation [20,21,22].

The beneficial effects of plant phenolic compounds on the improvement of inflammatory disorders have been reported in both in vitro and in vivo studies [23,24,25]. Methyl p-coumarate (methyl p-hydroxycinnamate) (MH), an esterified derivative of p-coumaric acid, was discovered in the flower of the medicinal plant *Trixis michuacana var longifolia* (D. Dow) C. [26], the roots and stems of *Comptonia peregrine* [27], the bark of *Melicope latifolia* [28], the leaves of Hainan *Morinda citrifolia* (Noni) [29], the fruit of Jujube (*Ziziphus jujuba* Mill.) [30], and in aloe vera [31]. Zhang et al. reported that MH has antibacterial properties [29], and Kubo et al. showed that MH has a suppressive effect on the formation of melanin in murine melanoma cell line B16 [32]. In addition, the experimental results from Lee et al. demonstrated that MH exerts an anti-inflammatory effect by suppressing the secretion of NO and the expression of iNOS in lipopolysaccharide (LPS)-stimulated RAW264.7 macrophages, and it exerts protective effects in mice with LPS-induced acute respiratory distress syndrome (ARDS) by inhibiting the secretion of TNF-α/IL-1β and the expression of iNOS [33,34]. However, to date and to the best of our knowledge, it has not been determined whether MH exerts an anti-asthmatic effect. Based on the inhibitory effect of MH on iNOS, which is an important factor in AA pathogenesis, protective effects of MH against AA are expected. Thus, the present study was undertaken to examine the protective effects of MH in an experimental model of AA.

## 2. Results

### 2.1. Suppressive Effects of MH against the PMA-Stimulated Inflammatory Response in A549 Cells

CytoX assay was used to assess the cytotoxicity of MH (5, 10, 25, 50, and 100 μM) in A549 cells or 10 nM PMA-stimulated A549 cells. The results showed that significant cell death did not occur until 100 μg/mL MH was administered (Figure 1a,b). Thus, doses were chosen for in vitro anti-inflammatory effects. ELISA results indicated that 10 nM PMA led to significant upregulation of IL-6 secretion (Figure 1c), whereas pretreatment with MH suppressed this upregulation in PMA-stimulated A549 cells. MH pretreatment inhibited the secretion of IL-8, MCP-1, and ICAM-1 in A549 cells following PMA administration (Figure 1d–f). In addition, MH suppressed the PMA-stimulated phosphorylation of IκBα and p65 in A549 cells (Figure 2a,b). It also exerted suppressive effects on NF-κB nuclear translocation in PMA-stimulated A549 cells (Figure 2c,d). In addition, MH exerted an inhibitory effect not only against c-Jun/c-Fos phosphorylation but also c-Jun/c-Fos nuclear translocation in PMA-stimulated A549 cells (Figure 2e–h). Collectively, the results from these in vitro studies confirmed that MH exerts anti-inflammatory effects in PMA-stimulated A549 cells by suppressing the secretion of cytokine/chemokine/adhesion molecules and the activation of NF-κB/AP-1.

### 2.2. Suppressive Effects of MH on Adhesion of Eosinophil and Airway Epithelial Cells

The infiltration of eosinophils into the airway epithelium is an important event in asthma development [35]. It has been proven that the promotion of ICAM-1 secretion in airway epithelial cells causes the recruitment of eosinophils [4,15]. Here, the inhibitory ability of MH against ICAM-1 secretion was notable in activated A549 airway epithelial cells (Figure 1f). Thus, we examined whether MH could affect the adhesion of airway epithelial cells and eosinophils. As shown in Figure 3a,b, the increase in fluorescence intensity that indicates the adhesion of eosinophil cell line EOL-1 and A549 cells was remarkably reversed by MH pretreatment.

### 2.3. Suppressive Effects of MH against the LPS-Induced Inflammatory Response in RAW264.7 Cells

Previous observations indicated the protective properties of MH in the LPS-stimulated inflammatory response in RAW264.7 macrophages by showing its suppressive effect on the secretion of NO/PGE2 and the activation of NF-κB [33]. However, the inhibitory ability of MH on cytokines and chemokines, such as IL-6 and MCP-1, which are important factors in AA development [6,7], was not confirmed in that study. Interestingly, the effects of MH on the suppression of molecules (TNF-α, IL-1β, IL-6, and MCP-1) were notable in LPS-stimulated RAW264.7 cells (Figure 4a–d).

### 2.4. Inhibitory Effects of MH on Recruitment of Immune Cells in Mice with AA

Based on in vitro anti-inflammatory ability, we examined the ameliorative ability of MH on airway inflammation in OVA-induced mice with AA (Figure 5). It is well -known that the recruitment of immune cells (eosinophils and macrophages) accelerates the inflammatory response in the airway [9]; therefore, the suppressive ability of MH on eosinophil/macrophage influx was examined. A significant upregulation in these cells was discovered in the BALF of the AA group compared to the NC group through Diff-Quik staining and cell counting (Figure 6a,b). However, a notable decrease in these cells was confirmed in the AA group with a 5 mg/kg administration of MH. Interestingly, its ability was comparable to that of a 1 mg/kg administration of DEX, which was used as a positive control.

### 2.5. Suppressive Effects of MH on Secretion of Th2 Cytokines, MCP-1, and IgE in Mice with AA

Next, we focused on MH suppression of Th2 cytokines, MCP-1, and IgE. ELISA results showed the remarkable upregulation of Th2 cytokines (IL−4, −5, and −13) in BALF of mice with AA (Figure 7a–c). This upregulation was significantly decreased in mice with AA and MH administration. Based on the role of the key chemokine, MCP-1, in eosinophil/macrophage infiltration [36], the inhibitory effect of MH on this molecule was examined. MH administration inhibited the increased MCP-1 level in BALF of the AA group (Figure 7d). In addition, MH resulted in the downregulation of OVA-specific IgE in the serum of mice with AA (Figure 7e). Generally, its inhibitory ability against IL-4, IL-5, MCP-1, and IgE secretion was similar to that of DEX.

### 2.6. Suppressive Effects of MH on Immune Cell Influx and Mucus Secretion in Mice with AA

To detect immune cell influx in lung tissue, we performed H&E staining. The results indicated an increased influx of immune cells around the airway in the AA group (Figure 8a). This trend was effectively ameliorated in both the MH- and DEX-treated groups. To confirm the secretion level of mucus in the lungs, PAS staining was performed. As shown in Figure 8b, the results indicated that mucus secretion was markedly increased in the airway epithelium of mice with AA (Figure 8b). However, this increase was attenuated with the administration of MH or DEX.

### 2.7. Suppressive Effects of MH on iNOS and COX-2 Expression in Mice with AA

The expression changes of iNOS and COX-2 in the lungs of mice were determined using Western blotting. As shown in Figure 9a,b, an increase in iNOS expression was observed in the lung tissue lysates of mice with AA compared to the control group, and this increase was suppressed in the AA group with MH administration. Subsequently, the expression level of COX-2 was examined, and the results revealed that the elevated expression of COX-2 in the lung tissue lysates of the AA group was reduced with MH administration. The effect of MH (5 mg/kg) on the suppression of these molecules was comparable to that of DEX (1 mg/kg).

### 2.8. Effects of MH on NF-κB Inactivation in Mice with AA

As shown in Figure 9c,d, increased p65 phosphorylation was confirmed in the lungs of the AA group, whereas this tendency was reversed by MH administration. In addition, MH had a reductive effect on IκBα phosphorylation in the AA group (Figure 9c,d). The ability of MH to suppress NF-κB and IκBα activation was similar to that of DEX.

## 3. Discussion

The airway epithelium is the main source of inflammatory-associated molecules [37], and airway epithelial cell-derived IL-6 affects T helper cell (Th cell)-derived cytokine production [5]. The upregulation of IL-8 has been reported in the airway epithelium of asthma patients [38,39]. Bronchial epithelial cell-derived MCP-1 is known to induce bronchial remodeling by affecting monocyte/macrophage recruitment against allergen exposure in subjects with AA [40]. In addition, airway epithelial cell-derived ICAM-1 is closely associated with the promotion of airway inflammation in subjects with AA through eosinophil recruitment [4,15]. Macrophages isolated from patients with asthma generate more IL-6 than those in patients without asthma [41]. Macrophages are well-known producers of MCP-1, and the amelioration of airway inflammation was related to MCP-1 suppression in an experimental mouse model of OVA-induced AA [16,42]. Collectively, the inhibition of airway epithelial cell- and macrophage-released molecules may ameliorate airway inflammation in subjects with AA. In the present study, MH notably inhibited not only IL-6/IL-8/MCP-1/ICAM-1 secretion in activated-airway epithelial cells but also airway cell/eosinophil adhesion. Furthermore, MH decreased the secretion of TNF-α, IL-1β, IL-6, and MCP-1 in activated macrophages. Since these results showed that MH has anti-inflammatory effects on both airway epithelial cells and macrophages, we evaluated its protective effect against airway inflammation in an AA animal model.

Cumulative evidence indicates a pivotal role of Th2 cytokines, MCP-1, and IgE in the development of AA. IL-4 causes Th2 cell differentiation and macrophage/B-cell activation [43]. IL-5 promotes eosinophilic inflammation in the pathogenesis of asthma [44]. IL-4 and -5 lead to MCP-1 upregulation in bronchial epithelial cells by affecting macrophage recruitment [40,45]. IL-13 affects not only B-cell activation, IgE production, and mast cell activation but also mucus secretion in subjects with AA [46]. In addition, modulating Th2 cytokines and IgE has been suggested as a therapeutic approach in asthma therapy [47,48,49]. This information highlights the importance of Th2 cytokines and IgE regulation in subjects with AA. In this study, MH exerted a regulatory effect on Th2 cytokines/IgE secretion, as well as immune cell recruitment, in mice with AA. Similar to the results from an in vitro study, the regulatory ability of MH on MCP-1 was discovered in an in vivo study. Our results indicated that MH may ameliorate asthma symptoms by modulating Th2 cytokines, MCP-1, and IgE.

As mentioned earlier, the abnormal production of mucus can obstruct airflow [10,11], and IL-13 affects goblet cell hyperplasia and mucus production [50]. As the ability of MH to suppress IL-13 was confirmed in an in vivo study, its suppressive ability in mucus secretion was also examined. Interestingly, its ability was similar to that of the positive control of DEX. These results reflect that MH has a regulatory effect on both IL-13 secretion and mucus production, suggesting that MH may improve the airflow obstruction in subjects with AA. Further confirmation of the inhibitory ability of MH on another factor affecting mucus production could highlight the anti-asthmatic effect of MH.

The upregulation of various molecules, including cytokines, chemokines, and adhesion molecules, has been reported in the pathogenesis of asthma [36,51,52] and in experimental models of inflammatory lung diseases, including AA [53,54,55]. Accumulated evidence has indicated that the activation of NF-κB plays a pivotal role in the expression of inflammatory molecules in lung epithelial cells and macrophages [5,13,15,56,57]. Chauhan et al. suggested that a reduction in NF-κB activation can ameliorate OVA-induced airway inflammation, leading to the suppression of inflammatory molecules [15,18,58]. Thus, targeting NF-κB pathways could be a promising therapeutic approach for AA. The in vitro and in vivo results from this study revealed that MH has an inhibitory ability against various molecules and NF-κB activation. Therefore, these results reflect that the protective effects of MH against airway inflammation in experimental asthma may be associated with the modulating ability of MH against NF-κB activation. However, further studies on the mechanisms of asthma improvement are needed before it can be developed as an adjuvant.

The information obtained from in vitro and in vivo studies has shown the biological properties of natural compounds from plants. Their regulatory effects against inflammatory molecules and NF-κB activation have been proposed for the treatment of inflammatory lung diseases, including AA [15,53,59,60]. Interestingly, in this study, the plant phenolic compound MH inhibited the production of various molecules and the activation of NF-κB in both in vitro and in vivo studies of experimental asthma, indicating the potential of natural compounds for asthma therapy. Compared with the anti-asthma effect of the existing natural compounds (40 or 200 mg/kg) [59,60], MH showed a superior effect at a low dose (5 mg/kg). Therefore, it is thought to be advantageous for development as an anti-asthma adjuvant.

## 4. Material and Methods

### 4.1. Reagents and Cell Culture

Methyl p-coumarate (methyl p-hydroxycinnamate) (MH) was obtained from SynQuest Laboratories, Inc. Airway epithelial cell line A549 and macrophage cell line RAW264.7 were obtained from the American Type Cell Culture (ATCC, Rockville, MD, USA), and were respectively grown in RPMI 1640 (contains 100 units/mL of penicillin in 100 μg/mL) and DMEM (contains 1% antibiotic antimycotic solution), supplemented with 10% FBS, at 37 °C in a CO_2_ incubator.

To determine the cell viability, A549 cells were placed into 96-well plates at an average density of 2 × 10^4^ cells per well and were maintained with MH (5–100 μg/mL) for 1 h. Then, the cells were incubated with 10 nM PMA or without PMA for 24 h. Subsequently, cell viability was measured in triplicate using a CytoX cell viability kit (LPS Solution, Inc., Daejeon, Korea).

To analyze the secretion levels of inflammatory molecules from A549 and RAW264.7 cells, A549 cells were placed in 96-well plates at an average density of 0.5 × 10^5^ cells per well, maintained with MH for 1 h, and incubated with PMA (10 nM) for 18 h. RAW264.7 cells were placed in 96-well plates at an average density of 0.5 × 10^5^ cells per well, maintained with MH for 1 h, and incubated with 500 ng/mL LPS for 18 h. Subsequently, the culture supernatants were assayed for cytokine, chemokine, and adhesion molecules using ELISA kits (BD Biosciences, Franklin Lakes, NJ, USA and R&D Systems, Inc., Minneapolis, MN, USA).

### 4.2. Nuclear and Cytoplasmic Extraction

The isolation of the nuclear and cytoplasmic fractions was performed based on previous protocols [61]. In brief, A549 cells were moved to a 60 mm cell culture dish at an average density of 2.5 × 10^5^ cells per dish and maintained with MH (10–100 μM) for 1 h. Subsequently, the cells were incubated with 10 nM PMA for 1 h. Finally, cell harvesting was performed using mechanical scraping, and the isolation of the nuclear and cytoplasmic fractions was performed using a NucBuster^TM^ Protein Extraction Kit (cat. no. 71183, Merck, Darmstadt, Germany).

### 4.3. Cell Adhesion Assays

Cell adhesion assays were used to determine the inhibitory ability of MH against the adhesion of airway cells and eosinophils based on previous protocols [15]. In brief, A549 cells were placed in 96-well plates at an average density of 1 × 10^4^ cells per well, maintained with MH for 1 h, and subsequently incubated with PMA for 5 h. Eosinophilic leukemia EOL-1 cells stained with calcein AM (cat. no. 4892-010-02, Bio-Techne Corp, Minneapolis, MN, USA), a cell-permeable dye that has been used for cell adhesion research [4], were placed at an average density of 5 × 10^4^ cells per well in 96-well plates containing A549 cells. After 1 h, the co-cultured cells were washed with PBS, and the degree of adhesion was quantitatively analyzed through the fluorescence intensity, measured using a fluorescence microscope (Eclipse Ti-U, Nikon, Tokyo, Japan, 490/520 nm).

### 4.4. Establishment of the OVA-Induced AA mouse model

Female BALB/c mice (6 weeks old, Koatech Co., Ltd., Pyeongtaek, Korea) were divided into four groups (*n* = 6 per group) as follows: (i) NC, normal control group; (ii) OVA, ovalbumin/alum-sensitized group; (iii) DEX, OVA + dexamethasone (DEX, 1 mg/kg)-treated group; (iv) MH, OVA + methyl p-hydroxycinnamate (MH, 5 mg/kg)-treated group.

The experimental AA mouse model was developed based on previous protocols [21]. In brief, the intraperitoneal injection (i.p.) of the OVA/alum mixture was performed on BALB/c mice on days 0 and 7. They were exposed to 1% OVA for 1 h daily on days 11 to 13. The oral administration (p.o.) of 5 mg/kg MH or 1 mg/kg DEX (positive control) was performed on days 9 to 13.

### 4.5. Immune Cell Count and ELISA

BALF collection to detect the immune cell count and Th2 cytokine production was performed under anesthesia, induced by the mixture of 30 mg/kg Zoletil and 5 mg/kg xylazine (i.p.) on day 15, as previously described [34]. Diff-Quik staining was performed using BALF cells to distinguish the morphology of immune cells, and the immune cells, such as eosinophils and macrophages, were counted using a light microscope (magnification, ×400) [62]. BALF supernatant was assayed for Th2 cytokines/MCP-1, and serum was assayed for IgE using ELISA kits (R&D Systems, Inc., Minneapolis, MN, USA and Biolegend, Inc., San Diego, CA, USA).

### 4.6. Western Blot Analysis

The lysates of cell culture and mouse lungs were prepared in a lysis buffer in the presence of protease/phosphatase inhibitors, and BCA protein quantification was then performed. Denatured protein samples were loaded on 10–15% SDS-PAGE gels and transferred to PVDF membranes. After blocking the membranes with 5% skimmed milk in TBST, the membranes were incubated with the following primary antibodies: phosphorylated (p)-IκBα (cat. no. 9246), p-NF-κB p65 (3033), p-c-Jun (32740), c-Jun (9165), p-c-Fos (5348), and c-Fos (2250) (all 1:1000; Cell Signaling Technology, Inc., Danvers, MA, USA); IκBα (cat. no. MA5-15132) and β-actin (MA5-15739) (both 1:1000; Invitrogen; Thermo Fisher Scientific, Inc., Rockford, IL, USA); NF-κB p65 (cat. no. sc-8008), iNOS (sc-651), COX-2 (sc-1747), and Lamin A/C (sc-376248) (all 1:1000; Santa Cruz Biotechnology, Inc., CA, USA). Then, the membranes were maintained with the corresponding secondary antibodies, washed with PBS, and exposed to ECL solution to visualize each band.

### 4.7. Histological Analysis

The lung tissues were washed, fixed with 10% formalin, and finally embedded in paraffin. The paraffin-embedded lung tissues were then sectioned at a thickness of 4 μm using a rotary microtome and stained with H&E and PAS staining solution [63,64].

### 4.8. Ethics Statement

The procedures for the animal experiments were approved by the IACUC on 23 March 2022 of the Korea Research Institute of Bioscience and Biotechnology (KRIBB, Chungbuk, Korea; KRIBB-AEC-22074). The procedure was also performed in compliance with the NIH Guidelines for the Care and Use of Laboratory Animals, as well as the Korean national laws for animal welfare.

### 4.9. Statistical Analysis

Values are expressed as the mean ± SD ≥ 3 independent experiments. One-way ANOVA followed by Tukey’s multiple comparison test were performed to analyze the differences between groups using SPSS 20.0 software (IBM Corp, Armonk, NY, USA). *p* < 0.05 was considered to indicate a statistically significant difference.

## 5. Conclusions

The modulatory effects of MH on airway cell/macrophage-derived molecules, airway cell/eosinophil adhesion, Th2 cytokine/MCP-1/IgE secretion, eosinophil/macrophage recruitment, mucus production, and iNOS/COX-2 expression were, in general, significant. These effects were accompanied by the suppression of NF-κB activation. These results demonstrate the anti-inflammatory effects of MH on airway inflammation and suggest that these effects may be based on NF-κB inactivation. Thus, it is suggested that MH may hold promise as an adjuvant in the treatment of AA. Further experiments confirming whether MH acts directly on the proteins may clarify the effect and mechanism of MH.

## Figures and Tables

**Figure 1 ijms-23-14909-f001:**
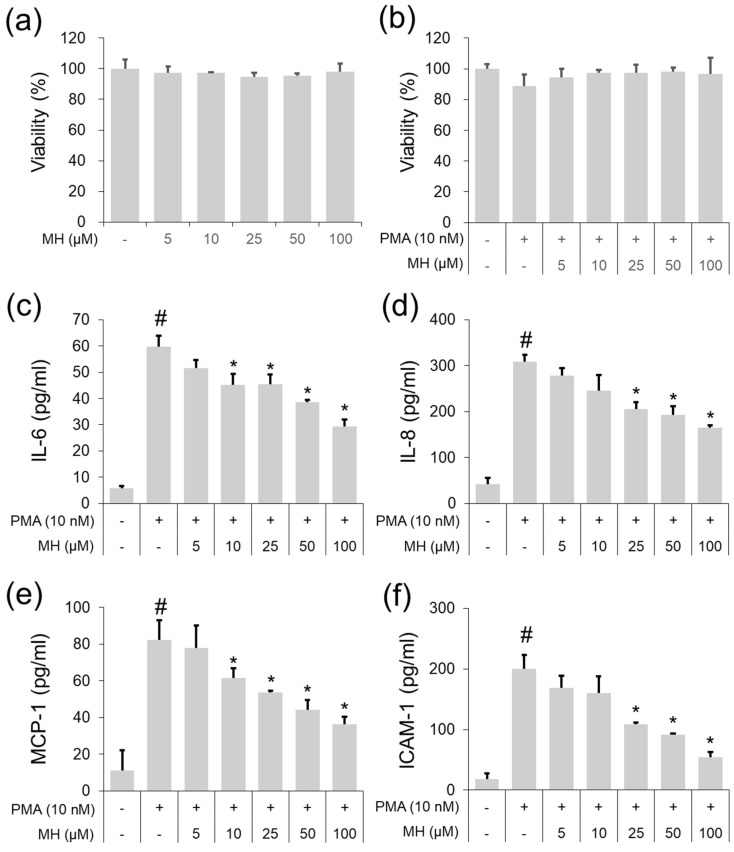
Effects of MH on secretion of cytokines, chemokines, and adhesion molecules in PMA−stimulated A549 cells. Cells were treated with MH (5, 10, 25, 50, and 100 μM) 1 h prior to incubation with 10 nM PMA for 18 h. (**a**,**b**) Cell viability was evaluated using CytoX assay. (**c**–**f**) The levels of IL−6, IL−8, MCP−1, and ICAM−1 were detected using ELISA kits. Data are expressed as the mean ± SD (*n* = 3) (^#^
*p* < 0.05 for comparison with control; ** p* < 0.05 for comparison with 10 nM PMA).

**Figure 2 ijms-23-14909-f002:**
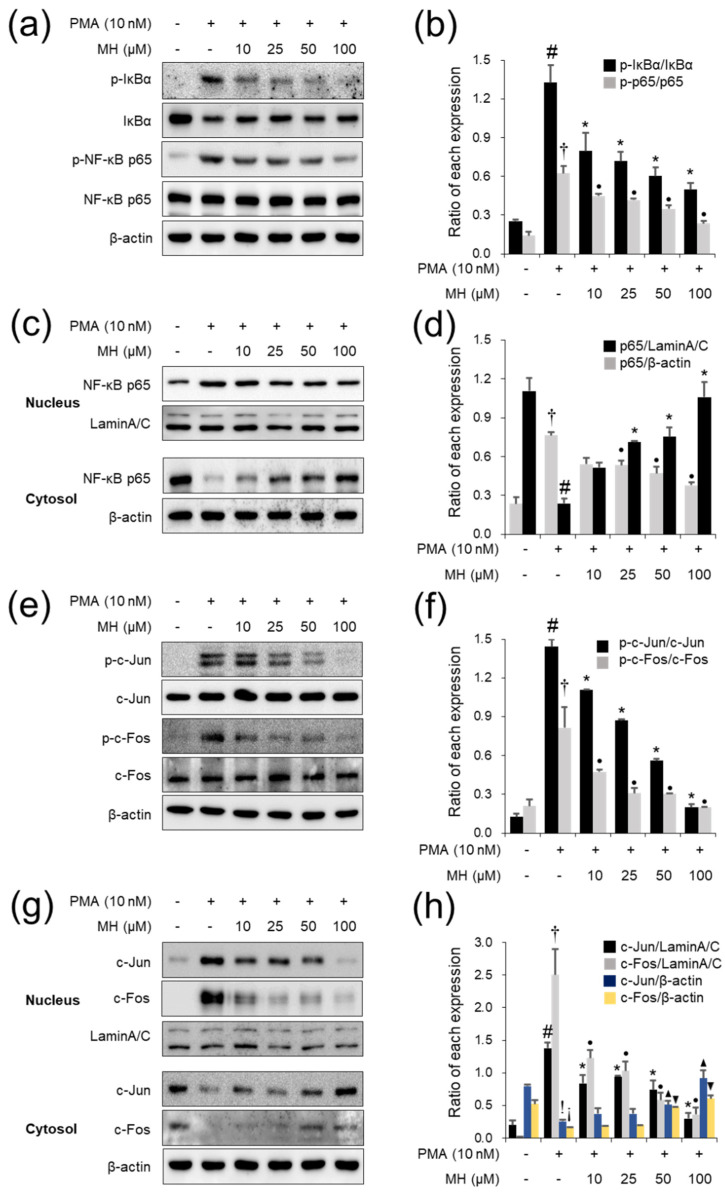
Effects of MH on activation of NF−κB and AP−1 in PMA−stimulated A549 cells. Cells were treated with MH (10, 25, 50, and 100 μM) 1 h prior to incubation with 10 nM PMA for 1 h. (**a**,**b**) The levels of p−IκBα and p−NF-κB p65 in whole cell lysate were detected using Western blotting. (**c**,**d**) The levels of NF−κB p65 in cytoplasmic cell fraction lysate and nuclear cell fraction lysate were detected using Western blotting. (**e**,**f**) The levels of p−c−Jun and p−c−Fos in whole cell lysate were detected using Western blotting. (**g**,**h**) The levels of c−Jun and c−Fos in cytoplasmic cell fraction lysate and nuclear cell fraction lysate were detected using Western blotting. Data are expressed as the mean ± SD (*n* = 3) (^#,†,!,¡^
*p* < 0.05 for comparison with control; *^,●,▲,▼^
*p* < 0.05 for comparison with 10 nM PMA).

**Figure 3 ijms-23-14909-f003:**
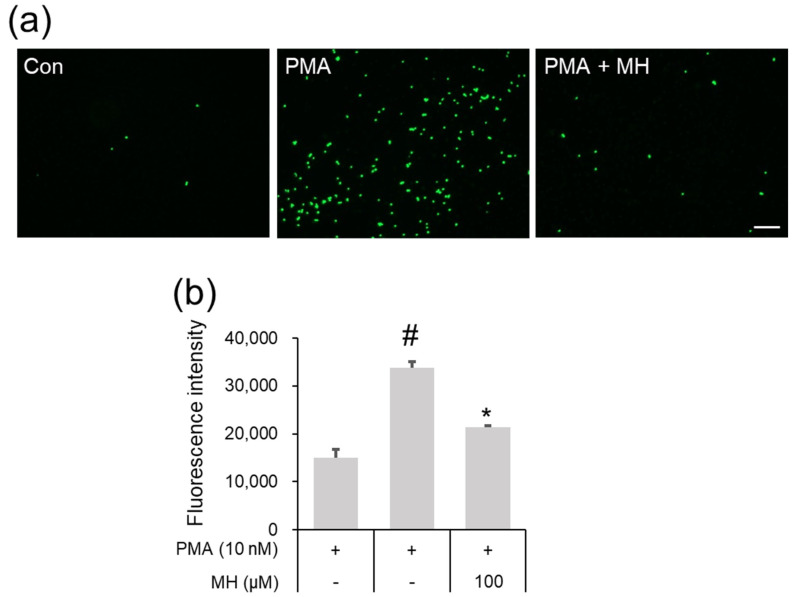
Effect of MH on the adhesion of A549 and EOL−1 cells. A549 cells were treated with 100 μM MH 1 h prior to incubation with 10 nM PMA for 5 h. Subsequently, EOL−1 cells stained with calcein AM were incubated with A549 cells for 1 h. (**a**) Green fluorescence, which indicated A549 and EOL−1 cell adhesion, was observed using fluorescence microscopy (magnification, ×100; scale bar, 100 μM). (**b**) The levels of EOL−1 cells adhering to A549 cells were quantified by detecting the fluorescence intensity. Data are expressed as the mean ± SD (*n* = 3) (*^#^ p* < 0.05 for comparison with control; ** p* < 0.05 for comparison with 10 nM PMA).

**Figure 4 ijms-23-14909-f004:**
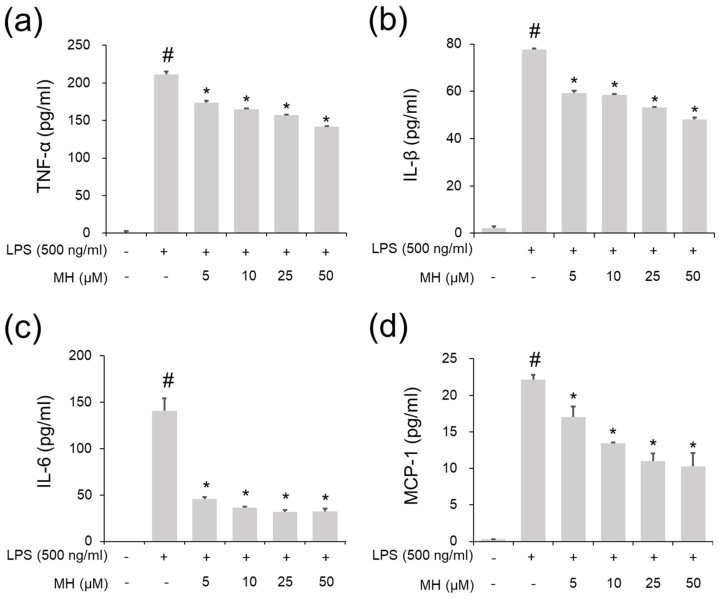
Effects of MH on secretion of cytokines and chemokines in LPS−stimulated RAW264.7 cells. Cells were treated with MH (5, 10, 25, 50, and 100 μM) 1 h prior to incubation with 500 ng/mL LPS for 18 h. (**a**–**d**) The levels of TNF−α, IL−1β, IL−6, and MCP−1 were detected using ELISA kits. Data are expressed as the mean ± SD (*n* = 3) (^#^
*p* < 0.05 for comparison with control; ** p* < 0.05 for comparison with 500 ng/mL LPS).

**Figure 5 ijms-23-14909-f005:**
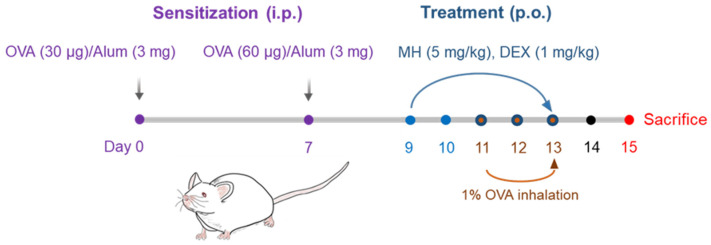
The establishment of the mice with AA and the administration of MH. BALB/c mice were randomly divided into four groups, sensitized with OVA/alum mixture on days 0 and 7, and exposed to OVA inhalation on days 11–13. Oral administration of MH or DEX was performed on days 9–13. On day 15, the mice were sacrificed, and the collection of BALF, serum, and lungs was performed for analysis.

**Figure 6 ijms-23-14909-f006:**
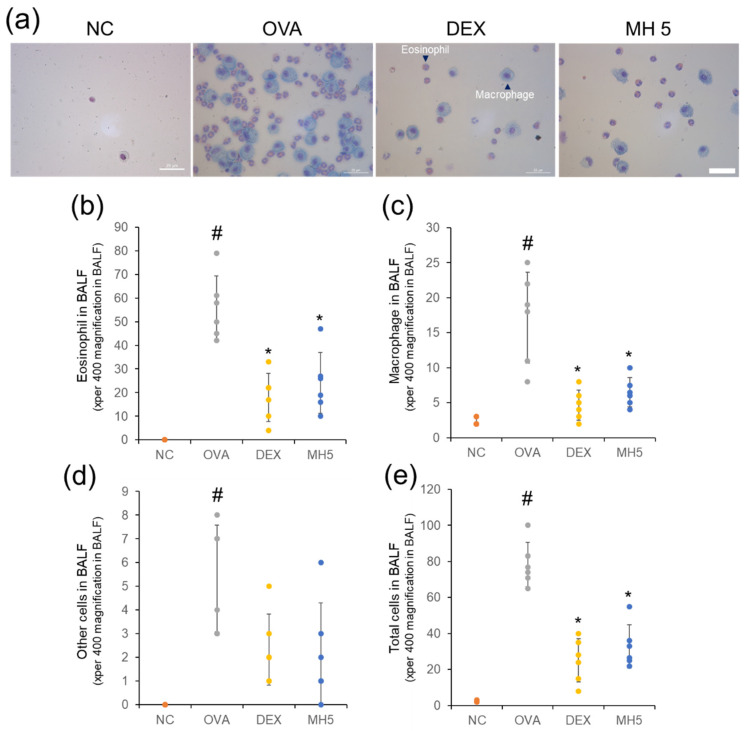
Effects of MH on recruitment of eosinophils and macrophages in mice with AA. (**a**) Microscope images of immune cells (magnification, ×400; scale bar, 25 μm). (**b**–**e**) The numbers of immune cells in BALF were determined using cell counting. NC: normal control mice; OVA: OVA-sensitized/challenged mice; DEX: 1 mg/kg DEX-treated OVA mice; and MH 5: 5 mg/kg MH-treated OVA mice group. Data are expressed as the mean ± SD (*n* = 6) (^#^
*p* < 0.05 for comparison with normal control; ** p* < 0.05 for comparison with OVA group).

**Figure 7 ijms-23-14909-f007:**
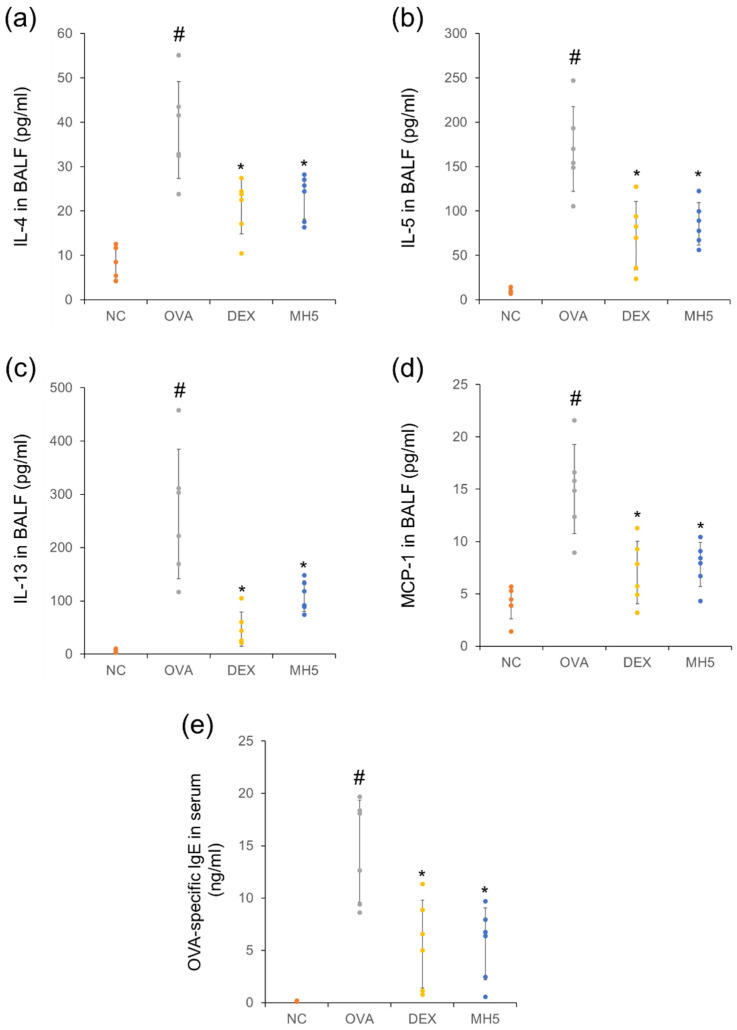
Effects of MH on the production of Th2 cytokines, MCP−1, and IgE in mice with AA. (**a**–**d**) The levels of Th2 cytokines (IL−4, −5, and −13) and MCP−1 in BALF were determined using ELISA assays. (**e**) The level of OVA−specific IgE in the serum was determined using ELISA assays. Data are expressed as the mean ± SD (*n* = 6) (^#^
*p* < 0.05 for comparison with normal control; ** p* < 0.05 for comparison with OVA group).

**Figure 8 ijms-23-14909-f008:**
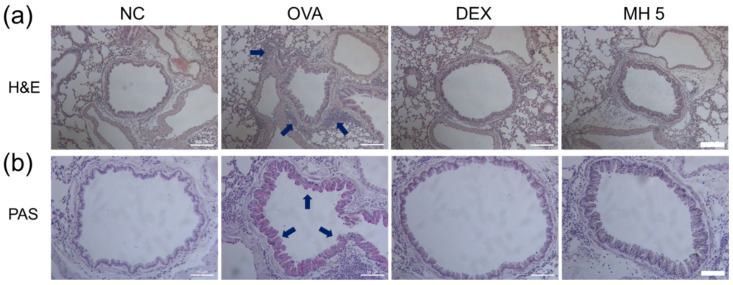
Effects of MH on immune cell influx and mucus generation in mice with AA. (**a**) The existence of immune cells surrounding the airway was confirmed using H&E staining (magnification, ×100; scale bar, 100 μM). (**b**) Airway mucus was detected using PAS staining (magnification, ×200; scale bar, 50 μM).

**Figure 9 ijms-23-14909-f009:**
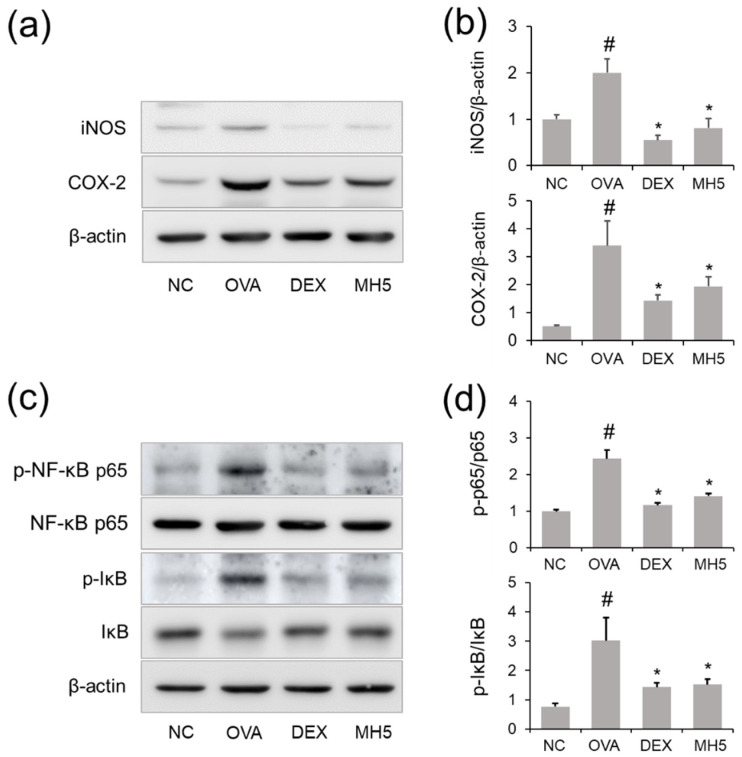
Effects of MH on iNOS/COX−2 expression and NF−κB activation in mice with AA. (**a**,**b**) The levels of iNOS and COX−2 in lung tissues were detected using Western blotting, and quantitative analysis was performed using ImageJ software. (**c**,**d**) The levels of p−NF−κB p65 and p−IκBα in lung tissues were detected using Western blotting, and quantitative analysis was performed using ImageJ software. Data are expressed as the mean ± SD (*n* = 3) (^#^
*p* < 0.05 for comparison with normal control; ** p* < 0.05 for comparison with OVA group).

## Data Availability

Not applicable.

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
