# Peer review of "Methyl P-Coumarate Ameliorates the Inflammatory Response in Activated-Airway Epithelial Cells and Mice with Allergic Asthma"

_ijms, 2022, doi:10.3390/ijms232314909_

Round 1

Reviewer 1 Report

In the present study, the authors evaluated the ameliorative effects of Methyl p-coumarate (methyl p-hydroxycinnamate) (MH), a natural compound found in  a variety of plants (MH) on airway inflammation by two experimental approaches: in vitro assays, using different kind of cells lines and a mice model of allergic asthma (AA).

The results showed a broad anti-inflammatory effect of MH. MH exerted anti-inflammatory activity on PMA-stimulated A549 airway epithelial cells by suppressing the secretion of IL-6, IL-8, MCP-1, and ICAM-1. In addition, MH exerted an inhibitory effect on NF-κB (p-NF-κB and p-IκB) and AP-1 (p-c-Fos and p-c-Jun) activation, but also on A549 cell and EOL-1 cell (eosinophil cell lines) adhesion. In LPS-stimulated RAW264.7 macrophages, MH had an inhibitory effect on IL-6 and MCP-1. The results from this in vivo study revealed that the increase in eosinophils/Th2 cytokines/MCP-1 in the bronchoalveolar lavage fluid (BALF) and of IgE in the serum of OVA-induced AA mice was effectively inhibited by MH administration. MH also exerted a reductive effect on the immune cell influx, mucus secretion, and iNOS/COX-2 expression in the lungs of AA mice. These effects of MH were accompanied by the inactivation of NF-κB. Collectively, the findings of the present study indicated that MH attenuates airway inflammation in mice with AA, suggesting its potential as an adjuvant in asthma therapy.

The hypothesis, objective and experimental design are interesting and well argue. Overall, the design is well defined.

However, in discussion should be included the limitations of this study and probably, to discuss the results compared to another natural compounds also described as with anti-inflammatory properties and potentially useful to asthma.

Also, there are several points that should be clarified, mainly related with the methods section.

In vitro assays:

How many assays were performed? The Figures 1-4 represent the inhibitory effect of HM in different experimental conditions but there is not any information about the number of experiments performed. Each figure should indicate the number of assays or duplicates that have been performed.

In vivo model:

How was determined the concentration of HM for animal models?

Each figure should indicate the number of assays or duplicates that have been performed.

Methods:

For all the methods, the manufacturer used should be indicated or the protocol explained. There are many methods described as “based on the manufacturer’s protocol” and there is not information about the manufacturer. It is impossible to replicate the assays presented in this work.

Specific details about the protocols used should be indicated.

Several examples:

Line 278: RAW264.7 were obtained from ATCC. What it is mean ATTC??

Lines 283-284: Then, the cells were incubated with 10 nM PMA or without PMA for 24 h. MTT assays were conducted based on the manufacturer’s protocol.

Lines 289-291: Subsequently, an ELISA assay was performed to determine cytokine, chemokine, and adhesion molecules using cell culture supernatant, based on the manufacturer’s instruction.

Lines 297-298: the isolation of the nuclear and cytoplasmic fractions was performed using a nuclear and cytoplasmic extraction kit (Cat, no 71183, Merck, Germany). This reference is NucBuster™ Protein Extraction Kit?? It is  for Nuclear and cytoplasmic extraction  or only to Nuclear??? It should be better to include the name of the kit used and not the reference.

Lines 304-306: Eosinophilic leukemia cell line EOL-1 cells stained with Calcein AM?? (manufacturer and at least to detail that is a green fluorochrome)  were placed into 96-well plates, which included A549 cells at an average density of 5 × 104 cells per well. After 1 h, the co-culture cells were washed with PBS, and the degree of adhesion was quantitatively analyzed through the fluorescence intensity measured by a fluorescence microscope. The conditions to detect the fluorescence of Calcein AM and the microscope used should be detailed.

Lines 327-328. An ELISA assay was performed to detect Th2 cytokines in BALF supernatant and IgE in serum was performed based on the manufacturer’s instructions???

Line 332. Denatured protein samples were loaded on SDS-PAGE gels. What percentage was used?

Reviewer 2 Report

In the manuscript entitled “Methyl p-Coumarate Ameliorates the Inflammatory Response in Activated-Airway Epithelial Cells and Allergic Asthma Mice”, the authors investigated the role of a natural product from plants, named Methyl p-coumarate (methyl p-hydroxycinnamate) (MH) in the inflammatory response. They used both in vitro and in vivo models in the study and found that MH has an anti-inflammatory effect on lung epithelial cells and can also inhibit OVA-induced immune cell influx, mucus secretion, and iNOS/COX-2 expression. The study is designed logically, and the conclusion is well supported by the results. However, several major concerns are observed as follows, which need to be addressed.

1.     Please display graphs as dot plots instead of bar graphs to show individual data points. Also, please show the animal number used for each experiment in figure legends.

2.     In Figure 4, it would be strengthened to show more cytokines or chemokines, such as TNF-α, and IL-1β.

3.     In Figure 6a, it is not convincing that there are no cells from BALF in the NC group. This result is biased and misleading. Please replace the image.

4.     Similarly, it is better to count the entire BALF cell number rather than only in images in Figure 6b.

Round 2

Reviewer 1 Report

The authors have correctly answered and clarify all the questions suggested. I agree with the present version.

Reviewer 2 Report

The authors were not able to address the comments adequately. All the graphs need to display as dot plots instead of bar graphs.
